# Enteroendocrine System and Gut Barrier in Metabolic Disorders

**DOI:** 10.3390/ijms23073732

**Published:** 2022-03-29

**Authors:** Céline Osinski, Dounia Moret, Karine Clément, Patricia Serradas, Agnès Ribeiro

**Affiliations:** 1NutriOmics, Research Group, INSERM, Nutrition and Obesities: Systemic Approaches, Sorbonne Université, F-75013 Paris, France; celine.osinski@sorbonne-universite.fr (C.O.); dounia.moret@sorbonne-universite.fr (D.M.); karine.clement2@gmail.com (K.C.); 2Nutrition Department, Pitié-Salpêtrière Hospital, Assistance Publique/Hôpitaux de Paris, F-75013 Paris, France

**Keywords:** enteroendocrine cells, cell lineage, intestinal hormones, obesity, type 2 diabetes, diets, microbiota, gut barrier integrity

## Abstract

With the continuous rise in the worldwide prevalence of obesity and type 2 diabetes, developing therapies regulating body weight and glycemia has become a matter of great concern. Among the current treatments, evidence now shows that the use of intestinal hormone analogs (e.g., GLP1 analogs and others) helps to control glycemia and reduces body weight. Indeed, intestinal endocrine cells produce a large variety of hormones regulating metabolism, including appetite, digestion, and glucose homeostasis. Herein, we discuss how the enteroendocrine system is affected by local environmental and metabolic signals. These signals include those arising from unbalanced diet, gut microbiota, and the host metabolic organs and their complex cross-talk with the intestinal barrier integrity.

## 1. Introduction

The intestinal epithelium supports major functions. The primary function is the role of the intestine barrier to protect the internal environment from the luminal content. The intestinal barrier is both physical and chemical. The physical barrier is achieved by the cell/cell tight junctions. The chemical barrier is brought by the antimicrobial peptides secreted by Paneth cells, such as defensins and lysozymes, and the mucus layer secreted by goblet cells. Immune cells present in the lamina propria also make the intestine an immune organ. One of the most obvious intestinal functions is nutrient absorption by enterocytes through numerous specialized transporters at the apical pole of the epithelial cells. Finally, the intestine is also an endocrine organ containing several enteroendocrine cells (EEC) secreting hormones that are involved in the control of energy homeostasis.

The homeostasis of the intestinal epithelium depends on a precise coordination of proliferation, migration, differentiation, and cell-death. The constant renewal of this epithelium occurs throughout life, every 3–5 days in humans [1], from intestinal stem cells (ISC) located in the crypt. ISC generate a population of transit-amplifying cells, making the crypt the proliferative compartment. After asymmetric division and proliferation, transit-amplifying cells differentiate and migrate along the crypto-villus axis making the villus the differentiated compartment. Epithelial cells are then eliminated into the intestinal lumen by anoikis, a form of apoptosis induced by cell and extracellular matrix disruption (Figure 1A).

The constant renewal capacity of the gut, thanks to stem cells, and programming differentiation of different cell types of the intestinal epithelium allow tissue homeostasis and fine adaptation to the environment, and in particular to the nutritional environment in physiological or pathophysiological conditions.

The present review focuses on the enteroendocrine system which includes the lineage of EEC, the different EEC types and their secreted hormones, and how this system adapts to physiological or pathophysiological environments (e.g., nutrients, bacterial metabolites) in metabolic diseases involving multi-organ cross-talk with the gut barrier.

## 2. The Enteroendocrine System

EEC are derived, as the other epithelial cell types, from ISC located at the bottom of crypts, intercalated between the Paneth cells and named crypt-based columnar (CBC) stem cells [2]. Signaling molecules that originate from mesenchymal cells located in the lamina propria but also from fibroblasts, immune cells, enteric neurons, and blood capillaries pathways, in the crypt compartment, referred as the niche, regulate ISC [3]. At least five signaling pathways, Wnt, BMP, Notch, EGF/EphB, and Hippo, have been involved in the ISC regulation.

Depending on the activation state of the Notch signaling pathway, the transient amplifying cells will differentiate into two lineages: absorptive lineage when the Notch signal is “ON” and secretory lineage when the Notch signal is “OFF” (Figure 1B). Enterocytes, the absorptive cells, represent 80 to 90% of the intestinal epithelial cells. At the apical and basolateral membranes, numerous transporters, receptors, and enzymes facilitate the selective transport of nutrients or metabolites from the intestinal lumen to the blood and lymph. Secretory cells include several cell types. Paneth cells, located at the bottom of the crypt, represent 3 to 7% of epithelial cells [4]. They are involved in the defense against pathogens present in the intestinal lumen by secreting antimicrobial peptides including lysozyme and defensins, and in ISC survival by producing essential molecules such as Wnt factor [5]. In the villus, three other secretory cell types are present. The tuft cells are few in number (0.4% of epithelial cells) and their role is still unclear, although an immune response against parasite infection was recently shown [6]. The mucus secreting cells (or goblet cells) are the most abundant secretory cells (16% of epithelial cells) [7]. They express and secrete mucins, forming a mucus layer that promotes the movement of the luminal content and isolates the epithelial monolayer from bacteria present in the intestinal lumen. The EEC, only 1% of the intestinal epithelial cells, produce and secrete around twenty enterohormones in response to various stimuli such as nutrients or bacterial metabolites, which makes the intestine the major endocrine system in humans.

### 2.1. Enteroendocrine Cell Types

By contrast to the gastric EEC, where they are “closed-type” with no contact with the lumen and not activated by external stimuli, EEC in intestine are “open-type” [8]. EEC in small intestine display a pyramidal shape with microvilli facing the intestinal lumen at the apical pole and a larger basal pole containing the secretory vesicles. EEC express at their apical pole, receptors that detect nutrients and metabolites and, in response to these stimuli, secrete several enterohormones that have paracrine, endocrine, or nervous effects.

Interestingly, for many years, each EEC subtype was considered to secrete only one enterohormone and was assigned a letter according to their specific synthesis of enterohormone: for instance, D cell for somatostatin (SST), I cell for cholecystokinin (CCK), K cell for Glucose-dependent Insulinotropic Polypeptide (GIP), and L cell for Glucagon-Like Peptide-1 and -2 (GLP-1 and GLP-2). However, the development of transgenic mice expressing fluorescent proteins under the control of hormone promoters led to revisiting this concept of “one EEC, one hormone”. Indeed, L cells are able to secrete both GLP-1 and PYY and these hormones are co-localized in human colon-derived L cells [9,10]. However, the co-existence of several hormones in the same secretory granules are still debated, also due to microscopy limitations. If GLP-1 and neurotensin, as well as PYY and neurotensin, are in distinct secretory granules in both the mouse ileum and in humans [11], the existence of distinct granules for GLP-1 and PYY is less certain. Nevertheless, Cho et al. reported that only 20% of granules contained both GLP-1 and PYY in the human jejunum [12].

Transcriptome analyses with the single cell RNA-seq were instrumental to confirm that one EEC can express multiple hormones. Furthermore, the combination of enterohormones differs depending on the intestinal segment studied [13,14] and the crypt–villus axis [15]. GIP secreting cells are present in the proximal part of the jejunum whereas the highest number of GLP-1 secreting cells is observed in the distal parts of the digestive system in both humans and mice [16]. While GLP-1 containing cells are predominantly present in crypts [14], in the villus EEC contain GLP-1, PYY, and neurotensin [11].

Many transcription factors are involved in the establishment of ECC lineage and act sequentially to generate the different EEC types with different hormone secretion potential. These transcription factors include NEUROG3, NEUROD1, PAX4 and PAX6, FOXA1 and FOXA2, ARX, and NKX2.2 (Figure 1C) [17,18,19,20,21,22,23]. A deeper description of the transcription factor network involved in the EEC lineage according to their coordinated temporal action was recently published by Gehart et al. [14]. In addition to genes encoding the transcription factors previously described to be involved in the lineage of EEC in mice, they identified six new genes: *Sox4*, *Rfx6*, *Tox3*, *Myt1*, *Runx1t1,* and *Zcchc12*.

### 2.2. Focus on GLP-1 and GIP Incretin Enterohormones

EEC secrete no less than twenty hormones (Figure 1C), and we herein focus on the incretin hormones GLP-1 and GIP that are extensively studied in metabolic diseases.

The incretin hormones GIP and GLP-1 potentiate insulin secretion in response to oral glucose intake [24]. These hormones contribute up to 50% of postprandial insulin secretion. Both GLP-1 and GIP are synthesized as pro-peptide, cleaved into hormones by pro-convertases, and are inactivated by the enzyme dipeptidyl peptidase-4 (DPP-4) [25]. This ubiquitous protein, also known as CD26, is present at the brush border of enterocytes and the plasma membrane of endothelial cells.

GIP is derived from pro-GIP cleaved by pro-hormone convertase 1/3 to GIP. The secretion of this hormone is stimulated in particular by dietary lipids but also by carbohydrates. The invalidation of the GIP receptor in mice protects these animals from high-fat diet-induced obesity [26]. Furthermore, the concentration of GIP secreted in response to a glucose bolus is positively correlated with body mass index [27]. In addition, GIP inhibits gastric acid secretion [25].

GLP-1 is encoded by the preproglucagon (*Gcg*) gene. *Gcg* is expressed in a subpopulation of EEC, in pancreatic α cells and in some neurons of the nucleus tractus solitarius [28]. After removal of the signal peptide, preproglucagon becomes proglucagon, which in turn leads to the formation of different peptides such as GLP-1 or glucagon under the action of tissue-specific proconvertases. In pancreatic α-cells, prohormone convertase 2 (PC2) predominates and leads to the production of the hyperglycemic hormone glucagon, as well as Glicentin-related pancreatic peptide (GRPP), Intervening peptide 1 (IP1), and Major Proglucagon Factor (MPF). In EEC and neurons of the nucleus tractus solitarius, the predominance of prohormone convertase 1/3 (PC1/3) results in the production of GLP-1, GLP-2, oxyntomodulin, glicentin, and Intervening peptide 2 (IP2) [28]. GLP-1 is produced as a 37 amino acid peptide after post-translational maturation, GLP-1(1-37). This form undergoes C-terminal amidation, forming GLP-1(1-36) amide. The action of PC1/3, by cleavage of six N-terminal amino acids, form GLP-1(7-37)- and GLP-1(7-36) amides, the latter being the predominant active form in blood. Both active forms of GLP-1 are rapidly inactivated by the action of the DPP-4 enzyme that cleaves active GLP-1 at the second N-terminal alanine. The produced GLP-1(9-37) and GLP-1(9-36) amide forms are thus inactive [29,30]. It is estimated that 25% of the active GLP-1 reaches the liver where it is also degraded by hepatic DPP-4 (40–50%) and 10–15% of active GLP-1 reaches the bloodstream [31]. Beside GLP-1 incretin’s effect on insulin secretion, GLP-1 exerts myriad metabolic effects, such as the control of food intake, gastric emptying and motility, increased muscle insulin sensitivity, and glucose uptake [32].

### 2.3. Mechanisms of Enterohormone Secretion

Enterohormone secretion is induced by all macronutrients (lipids, sugars, or proteins) and metabolites produced by intestinal bacteria (short chain fatty acids or SCFA, secondary bile acids). EEC contains at the apical membrane, numerous transporters, and detectors that trigger signaling pathways leading to hormone secretion in response to stimuli. These different signaling pathways result in an increase in intracellular Ca^2+^ concentration or in increase in cAMP. The transport of carbohydrates though SGLT1, GLUT5, and GLUT2 transporters leads to the production of ATP and the ATP-dependent potassium channels closure. The accumulation of potassium and sodium induces a plasma membrane depolarization. This change in membrane potential leads to the opening of voltage-dependent Ca^2+^ channels [16,33,34]. Membrane depolarization can also be induced by proton flux and PEPT1 activity in response to proteins.

Activation of protein transporter CASR, and G protein-coupled receptor (GPCR) such as sweet taste receptor (T1R2/T1R3) and lipid detectors (FFAR1, FFAR4, FFAR2, and FFAR3) induce the activation of phospholipase C beta (PLC) which cleaves Phosphatidylinositol-4,5-bisphosphate (PIP2) into Inositol 1,4,5-trisphosphate (IP3) and diacylglycerol [35]. IP3 binding to its receptor in the endoplasmic reticulum releases calcium from the endoplasmic reticulum.

GLP-1 secretion is “biphasic” with the first rapid and transient phase occurring between 1 and 6 min after stimulation and the second occurring between 7 and 12 min [36]. Rapid fusion of pre-existing vesicles close to the plasma membrane allows GLP-1 secretion within minutes while the second phase is due to fusion of neo-formed vesicles [37].

## 3. Models to Study Enteroendocrine Cells

With EEC being rare and scattered along the crypt–villus axis, models were developed allowing the study of EEC, from their transcriptomes to their secretory functions in response to stimuli. EEC lines and primary EEC are useful models, but in the last 10 years, intestinal organoid development, mimicking the intestinal epithelium and retaining characteristics from the intestinal region from which they originate, has provided a powerful model to study EEC function in response to physiological or pathological environment.

### 3.1. In Vitro Cell-Based Models

Currently, there are two different murine EEC lines, GLUTag and STC-1, and one human EEC line, NCI-H716. STC-1 secrete several enterohormones such as proglucagon, GLP-1, CCK, and strongly GIP [38,39,40]. In contrast to STC-1 cells, GLUTag cells are more differentiated, strongly express the GCG gene, and secrete GLP-1, GLP-2, and CCK [39,40]. The human cell line NCI-H716 also secretes GLP-1, GLP-2, and SST [40]. Methods were developed to obtain primary EEC from mice [34,41,42]. However, since EEC are scattered in the intestinal epithelium, this method has to be combined with selective markers of EEC such as fluorescent markers (e.g., transgenic mouse models whose proglucagon- or GIP-expressing cells are labeled with the fluorescent protein Venus) or with intra- (GLP-1, chromogranin A and Secretogranin 2) or extra-cellular markers (claudin-4) [34,43,44]. These tools provide access to the transcriptomic profile of EEC from different intestinal segments but also from the crypt or the villus [9,43,45,46]. Recently, our team developed an EEC cell sorting method from human and mice jejunum with a membrane marker, CD24, that allows different EEC type cell sorting without the need of cell permeabilization [47,48]. Primary GLP-1 producing cells, plated along with intestinal epithelial cell monolayer with a matrigel that mimics the intestinal basement membrane in composition and structure [41], also release GLP-1 [49,50] and are electrically excitable [34].

### 3.2. Enteroids

In the last decade, a more complex in vitro model containing the different intestinal epithelial cell types has been generated: a 3D “mini gut” structure with a crypt-like compartment containing stem cells and Paneth cells, and a villus-like compartment with enterocytes, mucus cells, EEC, and Tuft cells [51,52]. Enteroids are generated from adult donors CBC stem cells present in the crypt. By contrast, intestinal organoids are derived from embryonic stem cells or induced pluripotent stem cells (iPSCs). Below, we focus on enteroids that allow to study intestinal epithelial cells from donors and discuss their physiological properties.

The high capacity of intestinal renewal, thanks to the CBC stem cells, allows to cultivate intestinal crypts from adult mice in matrigel, an extracellular matrix enriched in laminin. This condition mimics the basement membrane and allows 3D culture [53]. The presence in the culture medium of growth factors similar to those present in the intestinal niche (Wnt, EGF, Noggin, and R-spondin), promotes the proliferation of CBC stem cells resulting in the formation of a spherical structure with buds. This sphere is bounded by an intestinal epithelial cell monolayer leaving a lumen inside and that recapitulate all the differentiated cell types from murine and human tissues [54]. Studies have shown that the enteroid identity from different parts of the intestine is kept with the maintenance of both enterohormone transcriptome and secretome [14,51,55,56,57,58].

Furthermore, these in vitro 3D models can be cultured with microorganisms for a better understanding of molecular mechanisms of interactions between microbiota and the host [52,59,60].

### 3.3. Exploring Enteroendocrine Cells in Enteroids

Despite the low EEC number in enteroids, it remains possible to study these cells by adding drugs or recombinant proteins to the culture medium. For example, the use of valproic acid (histone deacetylase inhibitor) and IWP-2 (Wnt pathway inhibitor) promote differentiation into enterocytes; DAPT (Notch pathway inhibitor) and CHIR99021 induce differentiation into Paneth cells [52]. DAPT can be used to promote mucus cell and EEC lineages in the presence [51], or absence of IWP-2 [61]. We have shown that treatment with DAPT makes possible the study of EEC in enteroids generated from patients with obesity and diabetes (Osinski C, unpublished data). Inhibitors of the Notch and/or Wnt pathway [51,61,62], of the ROCK pathway [63] or Ngn3 expressing lentivirus transduction [57,64] are used to increase the low EEC number in enteroids.

Several studies have investigated the impact of metabolites in enteroids secretion capacity, and in particular Short Chain Fatty Acids (SCFA), on EEC lineage. In murine and human duodenal enteroids, chromogranin A expression is increased in response to butyrate as early as 24 h of treatment [65]. A combination of acetate, propionate, and butyrate doubles the number of GLP-1-producing cells in murine and human enteroids [66]. Measurements of enterohormone secretion in response to various stimuli show that endocrine function is conserved in enteroids. In murine and human enteroids, GLP-1 and GIP secretions are amplified in response to glucose [42,62,67]. Other metabolites promote GLP-1 secretion such as di-peptide (Glycyl-Sarcosine) or a bile acid (deoxycholic acid or taurodeoxycholic acid) [42,67]. While Pearce et al. report an increase in PYY secretion, without mentioning GLP-1, in response to treatment with butyrate [65], Petersen et al. show that GLP-1 is secreted in response to a combination of SCFA (acetate, propionate, and butyrate) in both murine and human enteroids [66]. Transcriptome analysis of GLP-1 producing cells derived from human ileal-enteroids reveals the expression of genes involved in glucose sensing. These organoids secreted GLP-1 in response to glucose via the SGLT1 transporter [68].

An interesting question is how the enteroendocrine system is affected by local environmental and metabolic signals, including those arising from the diet, the gut microbiota, and from the host in pathological condition such as metabolic diseases. The next two chapters focus on enteroendocrine system in metabolic diseases, including crosstalk with gut microbiota and gut barrier.

## 4. Enteroendocrine System in Metabolic Disorders

Obesity is associated with metabolic alterations, including T2D, cardiovascular diseases and non-alcoholic fatty liver disease. The incidence of obesity and its co-morbidities has increased severely since the mid-twentieth century all over the world. Currently, the global average prevalence of obesity is 11% in men and 15% in women, and severe obesity affects 2% of men and 5% of women [69]. Obesity in children and adolescents has also increased in most countries over the past four decades [70]. Sedentary lifestyle, unbalanced diet, and polygenetic susceptibility are considered foundations for the obesity [71].

Obesity and T2D have detrimental effects on both gut hormone secretion and EEC lineage; however, the cause and consequences of their interplay need to be better understood. GLP-1 and PYY release in response to meals is impaired in obese patients [72,73,74,75,76,77]. The incretin effect, which enhances glucose-dependent insulin secretion in the postprandial state, is impaired in subjects with T2D, explained by combined effects of reduced GLP-1 secretion and impaired GIP action [77,78]. Nevertheless, the interplay between T2D state and GIP- and GLP-1-impaired secretions in human is not clear. Indeed, studies on GIP secretion during T2D are contradictory, showing either unaffected, increased, or even decreased secretion [79,80,81]. Young adults with obesity and a recent diagnosed T2D had an increased GLP-1 secretion when compared to obese patients without T2D [82]. This rise in GLP-1 could be transient and explained by the early stage of T2D. Another factor affecting the plasma GLP-1 level is the gastric emptying rate, which can be slower in patients with chronic T2D; e.g., slower gastric emptying indeed delays nutrient-induced GLP-1 secretion [83].

It is also speculated that metabolic diseases could alter the expression of transcription factors controlling EEC differentiation and thus EEC number, but with controversial results. In obese Zucker diabetic rats, an increased number of GLP-1 cells in the jejunum and ileum is due to the doubling of the intestinal surface [84]. Again, an increased number of GLP-1 cells is found in the duodenum of some obese patients with T2D [85]. Another study indicates similar GLP-1 cell number in the ileum of patients with or without diabetes [86]. A reduced EEC number was described in duodenum of subjects with morbid obesity [87]. Our team showed that transcriptomic profiling of EEC separates obese patients according to their diabetic status [47]. Furthermore, T2D was associated with an impaired GLP-1 cell differentiation leading to low GLP-1-cell density in human obesity. These mechanisms could account for the reduced plasma GLP-1 observed in metabolic disorders [47].

### 4.1. Diets and Enteroendocrine System

Although the diet modulates both EEC function and density, it is still debated whether it is the diet itself or the diet-induced metabolic disorders that affect the EEC and particularly GLP-1 secretion potential.

In rodents fed with sugar- or lipid-enriched diets, the balance between proliferation and differentiation of intestinal epithelial cells is altered [88,89]. A high-fat (HF) diet for 16 weeks induces a decrease in the number of GLP-1 cells in the colon, but not in the small intestine in the GLU-Venus (fluorescent GLP-1 cells) mouse model. In contrast, primary culture of intestinal cells from these mice showed a decreased GLP-1 secretion in response to glucose and peptone [46]. Another study highlighted that mice fed a HF diet for 16 weeks could be stratified into two groups depending on glucose tolerance status. The hyperglycemic group, but not the intolerant group, showed an increase GLP-1 cell number in the small intestine, whereas both groups showed an increase in GLP-1 secretion, compared with the control group [50]. These results are consistent with our study reporting an increased GLP-1 cell number in the jejunum and colon of hyperglycemic mice as early as 2 weeks of HF diet concomitant with an increased plasma GLP-1 concentration [90]. Furthermore, a positive association between jejunal GLP-1 cell density and fat consumption has been observed in individuals with severe obesity [90].

A study conducted in obese women following a Paleolithic diet based on vegetable proteins and oilseeds showed a weight loss after 6 months, more than women in the healthy control diet, based on Nordic Nutrition Recommendations [91]. Weight loss triggered an increase in postprandial GLP-1 levels and a further increase arose during weight maintenance. Postprandial GIP levels also increased after the Paleolithic diet [91], emphasizing again the importance of the diet in incretin release.

In EEC, carbohydrates and sweeteners may be detected by the sweet taste receptor. In the intestine, the activation of the sweet taste receptor activates a signaling pathway leading to GLP-1 secretion [92]. We showed that metabolic disorders in mice lead to altered gene expression of the sweet taste signaling pathway in intestine and contribute to impaired GLP-1 secretion [48]. After entero-gastro anastomosis procedure in mice, mimicking bypass surgery in humans, increased expression of gene encoding α-gustducin contributed to metabolic improvement [48].

### 4.2. Impact of Bariatric Surgery on Enteroendocrine System

Most understanding of the detrimental impact of metabolic diseases on gut hormones is provided by results obtained in patients with obesity in bariatric surgery programs, these surgical procedures being dedicated to subjects with severe or morbid obesity. After Roux-en-Y Gastric Bypass (RYGB) and Sleeve Gastrectomy (SG), weight loss is maximal between the first and second year [93]. Bariatric surgery reorganizes intestinal anatomy and shows, in clinical studies, a marked elevation of GLP-1 and PYY postprandial concentrations [94,95,96,97,98], probably contributing to postsurgical weight loss and improved glucose metabolism. RYGB indeed seems to induce major improvement of glucose metabolism and diabetes resolution in a large number of patients [99] even if the meta-analysis performed by Han et al. indicates that the remission of T2D is similar after SG or RYGB [100]. The mechanisms involved in these improvements are numerous.

The increased GLP-1 and PYY secretions in patients after bariatric surgery occurs quickly, even before major weight loss. Bariatric surgeries, especially RYGB, lead to a rearrangement of the alimentary circuit, resulting on the arrival of undigested food in the distal part of the intestine. This distal intestinal segment contains high number of GLP-1 and PYY cells supporting enterohormones secretion after bariatric surgery. Larraufie et al. showed that the enhanced postprandial GLP-1 release after bariatric surgery is mainly due to an altered flow of nutrients that stimulates more distal EEC rather than changing EEC characteristics or tissue hormones [101]. The absorption of glucose and proteins is strongly accelerated after an RYGB while it is only modestly accelerated following of SG [98]. The enhanced GLP-1 secretion observed after gastric bypass may be obtained by delaying carbohydrate digestion [102]. The massive influx of nutrients to the distal EEC after bariatric surgery cannot, however, totally explain the increase in plasma concentrations of enterohormones. Although more moderate, increased enterohormone levels are also observed after SG [87,103]. In addition, the secretion of more proximal enterohormones, such as GIP, is also impacted, although this remains less clear [96,98,103]. Thus, it is possible that the intestinal epithelium which is in constant renewal, undergoes remodeling following the rearrangement of the food circuit.

In humans, two studies indicated an increased density of GLP-1, GIP, and PYY cells [104,105], while another study showed that densities of chromogranin A and GLP-1 cells remain unchanged [106]. In rats, RYGB surgery does not have an impact on the EEC density [84,106,107]. Studies in rats undergoing SG indicate conflicting results regarding GLP-1 and GIP cell densities [106,108]. After metabolic improvement with entero-gastro anastomosis surgery in mice, the expression of *Gnat3* increased in the new alimentary tract and glucose-induced GLP-1 secretion was improved. Our data emphasize that metabolic disorders were associated with altered gene expression of sweet taste signaling in intestine. This could explain the impaired GLP-1 secretion that is partly rescued after metabolic improvement [48].

Other segments of the gastrointestinal tract can be altered after bariatric surgery. In the stomach, a hyperplasia of mucosecretory cells and a decrease in ghrelin mRNA levels in rats after RYGB or SG was shown [109]. More recently, an increase of GLP-1 cell number in the stomach of rats after RYGB or SG was described, with similar findings in the stomach of patients after RYGB [110]. In colon, an increased expression of several GPCR, involved in nutrient detection and enterohormone secretion, were described in mice after RYGB [111].

Altogether, this emphasizes that diets, metabolic diseases, and bariatric surgery deeply alter the enteroendocrine system, and notably incretin release. Moreover, the gut microbiota producing metabolites also modulate enteroendocrine system.

## 5. Microbiota, Enteroendocrine System, and Gut Barrier in Metabolic Disorders

The gut is colonized by trillions of bacteria that maintain symbiotic relationships with the host physiology. The gut microbiota metabolizes dietary- and host-derived factors to produce microbial metabolites, which are involved in many metabolic processes modulating energy and glucose homeostasis, gut barrier function, and systemic inflammation. Disruption of the equilibrium of host–microbiota interactions may contribute to obesity and related comorbidities as T2D and liver and cardiovascular disease [112,113].

Microbial metabolites are important actors of the links between the gut microbiome and the perturbed host biology in metabolic disorders. Metabolically beneficial metabolites include, in a non-exhaustive list, SCFA, bile acids, and vitamins, while potentially harmful metabolites include LPS, flagellin, and imidazole propionate [112,114] (Figure 2).

### 5.1. SCFA, Secondary Bile Acids, and GLP-2

The primary interface of microbiota–host interactions is the intestinal epithelium containing EEC. The gut microbiota plays an important role in EEC and their hormone release. Notably, the fermentation of the dietary fibers by gut microbiota in the distal part of the intestine produces SCFA, such as butyrate, propionate, and acetate. The SCFA influence the host metabolism in multiple manner by acting on GPCR, such as free fatty acid receptors (FFAR2 and FFAR3) expressed by EEC. Acetate and butyrate stimulate GLP-1 and PYY release with subsequent effects on the pancreas and on the brain, and possibly other tissues. Acetate may enhance fat storage by inducing secretion of ghrelin [71]. The level of SCFA-producing bacteria (e.g., butyrate producers) and SCFA is reduced in fecal samples of dysmetabolic mice and in humans with obesity and diabetes [112]. In rodents with metabolic disorders, supplementation with SCFA improves the metabolic phenotype by increasing energy expenditure, glucose tolerance, and homeostasis. In humans, SCFA administration stimulates the production of GLP-1 and PYY, leading to a decrease in weight gain [112,115].

Gut microbiota-induced SCFA production influences the differentiation of EEC and hormones synthesis by these cells. Several studies demonstrated that SCFA production, notably butyrate and propionate, associates with an increased number of GLP-1 producing cells, GLP-1 synthesis, and secretion in rodent colon [116,117,118,119]. In addition, in vitro SCFA promote GLP-1 cell differentiation and GLP-1 release in mouse and human enteroids [66] and stimulate GLP-1 and PYY secretion via their action on FFAR2 [117,118], resulting in improved glucose tolerance and in prevention of obesity development [118,119]. Germ-free mice exhibit a severely reduced SCFA levels, but an unexpectedly increased number of GLP-1 cells in the distal intestine and a higher GLP-1 plasma level, affecting the animal intestinal transit. The phenotype is rescued upon the colonization of the gastro-intestinal tract. This suggests that gut microbiota produced SFCA are not the sole regulator of GLP-1 cell activity. The authors suggest that the increased GLP-1 production is an adaptive response to altered homeostasis caused by the loss of the SCFA as an energy source, slowing intestinal transit and favoring nutrient absorption [120]. Recently, another study suggested that GLP-1 cell differentiation is modulated in a SCFA-Neurog3-Foxo1-dependent manner. The authors showed that SCFA promote FOXO1 transcription factor, which inhibits *Neurog3* gene expression and thus alter EEC differentiation. This mechanism may explain the higher GLP-1 cell number found in germ-free mice because of their reduced SCFA production [121]. The transcriptome of the germ-free GLP-1 producing cells is distinct from that of conventional mouse. The colonization of the gastro-intestinal tract of germ-free mice restores the GLP-1 producing cells transcriptome at very short term [122]. Overall, the gut microbiota and its metabolites, notably SCFA, partially influence the synthesis and secretion of enterohormones and also modulate the EEC number and their transcriptome. Mechanisms linking gut microbiota, in particular SCFA, and GLP-1-producing cells need further clarification.

Bile acids are synthetized in hepatocytes from cholesterol. Primary bile acids are converted by gut microbiota to secondary bile acids that act through the GPCR named TGR5 (GPR131). TGR5 signaling significantly activates GLP-1 secretion [123]. Gut dysbiosis affects the quality and quantity of secondary bile acids leading to defective activation of TGR5 and a reduced GLP-1 secretion [124]. Oral vancomycin is well known to reduce microbiota diversity, particularly the Firmicutes phyla. The decrease within the Firmicutes phyla was associated with less transformation of secondary bile acids and subsequent decreased insulin sensitivity in patients with metabolic syndrome [125,126]. These findings show that gut microbiota dysbiosis contributes to the defective transformation of bile acids and impacts glucose metabolism in humans [124], but the clarification of mechanisms needs further investigations.

Molecular links between enteroendocrine function and gut barrier are still debated and the literature on this subject is scarce. One example is provided by the study of GLP-2 enterohormone, which is involved in the control of epithelial cell proliferation and gut barrier integrity [127]. An increase in endogenous GLP-2 production was associated with improved barrier function via the recovery of tight junction protein expression and distribution in jejunum and colon [128]. Using a series of complementary approaches with specific modulation of the gut microbiota (antibiotics, prebiotics) or pharmacological inhibition or activation of the GLP-2 receptor, it was shown that the gut microbiota and the enterohormone GLP-2 both participate in gut barrier function and further associate with systemic and hepatic inflammation associated with obesity and T2D [128]. A recent study in obese mice treated with a phytonutrient as an anti-obesity agent showed beneficial effects on metabolism and inflammation parameters such as reduction in body weight gain, alleviated insulin resistance, decreased serum proinflammatory LPS, attenuated gut microbiota dysbiosis, and increased SCFA production. These beneficial effects were concomitant of an increased GLP-2 secretion and the restauration of gut barrier integrity [129] This evidence is consistent with the control of gut permeability and metabolic endotoxemia by EEC. Interestingly, the increase in endogenous GLP-2 has been associated with an increased number of GLP-1 producing cells in obese mice upon prebiotic treatment [130].

### 5.2. LPS, Flagellin, and Hyperglycemia

LPS are the major outer surface membrane components present in most Gram-negative bacteria [71]. LPS are released at bacterial cell death and are strong pro-inflammatory compounds that stimulate host immunity and promote inflammation. For instance, after HF diet in animals or in the endotoxemia state, they trigger local and systemic inflammation and insulin resistance [116]. Several chronic disorders characterized by a disturbance of the gut barrier, with imbalances in both intestinal and systemic immune responses, converge to cause insulin resistance, eventually leading to metabolic syndrome [71]. Long-term LPS subcutaneous infusion in mice recapitulates the pathological phenotype of HF diet mice: increased weight gain, insulin resistance, white adipose tissue inflammation, increased systemic LPS, and increased intestinal permeability [131].

Consuming different diets modulates the intestinal microbiota and its associated metabolic health. For instance, palm oil gavage in mice is proinflammatory and prompts a rapid disruption of the intestinal cell–cell junctions, an increased gut permeability, and inflammation before any significant weight gain [131,132]. HF diet can lead to changes in the gut microbiota that strongly reduces intestinal epithelium integrity due to the dysfunction of tight junction proteins, such as occludin and ZO-1 [133]. Consequently, these changes result in increased plasma levels of LPS. Circulating LPS activate Toll-like receptor 4 (TLR4) in systemic cells, resulting in chronic low-grade inflammation in different tissues [124].

TLR5 is a component of the gut immune system that helps to defend against infection. TLR5 is known to specifically sense and recognize flagellin, the major structural protein and the locomotion element of bacterial flagella [134]. TLR5 is expressed mostly on the basolateral pole of intestinal epithelial and detects whether bacteria have crossed the gut epithelium. Thus, TLR5 senses the composition and localization of the intestinal microbiota preventing diseases associated with intestinal inflammation in mice [135]. Abnormal increased TLR5 activation in response to flagellin might result in an impairment of gut barrier or the exacerbation of local chronic inflammation. Inappropriate flagellin recognition by TLR5 is also linked to modifications in the gut microbiota composition, abnormal adipose tissue metabolism and inflammation [136]. Mice genetically deficient in TLR5 exhibit hyperphagia, and develop low-grade inflammation and metabolic syndrome, including hyperlipidemia, hypertension, insulin resistance, and increased adiposity [137]. Mechanistic studies in rodents have also shown that hyperglycemia *per se* may increase intestinal barrier permeability through a GLUT2-dependent alteration of tight junction integrity, subsequently causing a leaky gut [138]. Moreover, in human obesity, increased jejunal permeability is linked to low-grade inflammation and T2D [139]

Translating these results in human studies is challenging. Nevertheless, recent indirect evidence confirms the interaction between the intestinal microbiota and metabolic diseases in humans. While the presence of bacteria had previously been suggested in the blood or within metabolic tissues, probably due to increased intestinal permeability, these features were recently confirmed by an independent group and showed associations with metabolic alterations [131].

## 6. Concluding Remarks

Although the enteroendocrine function is an important mechanism in regulating gut barrier, molecular links are debated. Literature on the links between enteroendocrine system in metabolic diseases and perturbed gut barrier function is scarce. Gut microbiota-derived metabolites have a central role in the physiology and physiopathology of metabolic disorders but links with EEC system and gut barrier need to be deciphered. Targeting enteroendocrine function and improvement of barrier integrity are considered as a relevant therapeutic approach to treat the intestinal and systemic inflammatory phenotype associated with obesity and T2D.

## Figures and Tables

**Figure 1 ijms-23-03732-f001:**
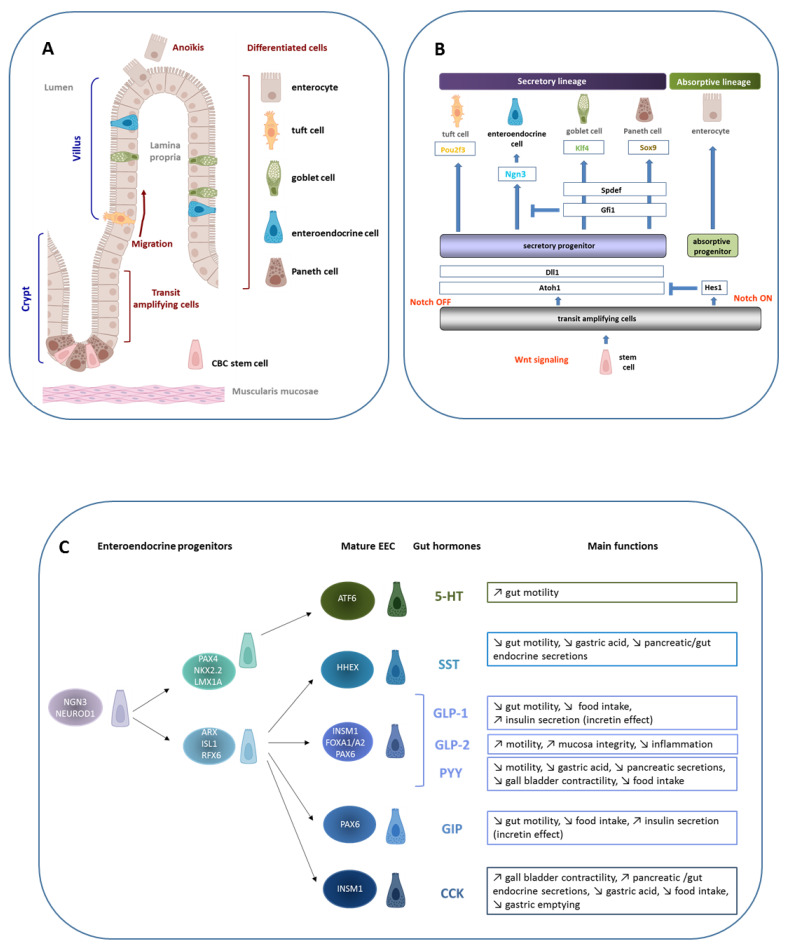
**Intestinal epithelium homeostasis, enteroendocrine cell lineage, and functions.** (**A**) The intestinal epithelium is composed of a single layer of cells lying on a mesenchymal tissue, the lamina propria, which contains connective tissue and vascular structure. In the small intestine, the epithelium is organized into villi that project in the intestinal lumen, with crypts which invaginate in the mucosa. This structural unit rests on a thin layer of smooth muscle, the muscularis mucosa. The intestinal epithelium is completely renewed in 3 days in mice and 5 days in humans from crypt-based columnar (CBC) stems cells located in the crypt bottom. After division, proliferating cells start to differentiate and migrate toward the villus tip where they are exfoliated after apoptosis, a process called anoïkis. Except for the Paneth cells that are located in the crypt bottom and intercalated between CBC stem cells, participating in the formation of the «niche», the other differentiated cells are located along the villus, enterocytes, tuft cells, goblet cells, and enteroendocrine cells (EEC). (**B**) Schematic demonstrating that under the Wnt signal, CBC stem cells proliferate and give rise to transit amplifying cells. Under the Notch signal, the transcription factor Hes1 is expressed and allows the progenitors to differentiate into enterocytes, the absorptive cells. When the Notch signal is OFF, Atoh1 is expressed and gives rise to the secretory lineage. The differentiation of secretory progenitors is under the control of specific transcription factors. The expression of Ngn3 leads to enteroendocrine progenitors. (**C**) A focus on the EEC linage shows its complexity. EEC lineage is under the control of a transcription factor network that acts sequentially leading to a collection of mature EEC characterized by the hormone they produce. Although EEC represent only 1% of the total intestinal epithelial cells, the large variety of hormones they secrete makes the intestine a major endocrine organ. Examples of gut hormones and their main biological functions are listed: serotonin (5-HT), somatostatin (SST), GLP-1 and GLP-2, PYY, GIP, and CCK.

**Figure 2 ijms-23-03732-f002:**
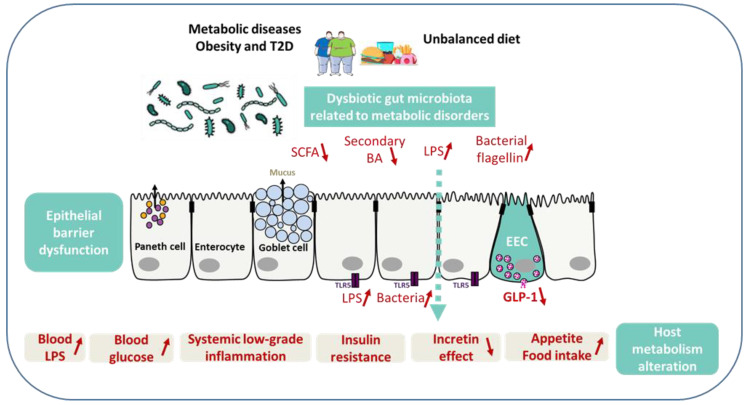
**Gut barrier, enteroendocrine cells, and metabolic signals related to metabolic diseases.** Unbalanced diet and dysbiotic gut microbiota related to metabolic disorders may result in dysfunction of intestinal barrier integrity, local and systemic inflammation, and reduced production of both SCFA and secondary bile acids, leading to less GLP-1 secretion by EEC. The reduced GLP-1 secretion leads in turn to impaired incretin effect, and an increase of blood glucose, appetite, and food intake. When in excess, LPS prompt intestinal and systemic inflammation and insulin resistance. TLR5, expressed mainly on the basolateral membrane of intestinal epithelial cells, identifies the bacterial locomotion component flagellin for uncovering whether bacteria have crossed the gut epithelia. Aberrantly elevated TLR5 activation might result in damage of the epithelial barrier integrity and inflammation.

## Data Availability

The data presented in this study are available on request from the corresponding author.

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
