# Peer review of "Enteroendocrine System and Gut Barrier in Metabolic Disorders"

_ijms, 2022, doi:10.3390/ijms23073732_

Round 1

Reviewer 1 Report

The review discusses the enteroendocrine system affection by local environmental and metabolic signals, due to intestinal hormone analogs use, in order to control glycemia and reduce body weight. The review contributes to the knowledge of intestinal endocrine cells production about the variety of hormones regulating metabolism, including appetite, digestion, and glucose homeostasis.

The review is covering completely the review topic and it also, covers the gap in knowledge.

The text is clear,comprehensive and of relevand to the field. There is no similar review published recently.

The cited references are not mostly within the last 5 years, but are appropriate. At my knowledge there are no omitted citations. The review does not include an abnormal number of self-citations.

The statements and conclusions are coherent and supported by the listed citations.

The figures are masterpieces and they properly show the data. Moreover, the figures are easy to interpret and understand.

Reviewer 2 Report

Osinski et al present a very organized and thorough review discussing the role of enteroendocrine cells (EECs) in health and metabolic disease. Their clear schematics are highly effective and notably aesthetic. The review introduces general EEC biology, the molecular cues that regulates their differentiation and function, key hormones, their mechanisms of action in target tissues and commonly used EEC models.

Subsequent sections focus on the role of EECs in the context of metabolic disease (obesity and T2D) and carefully highlight key literature in mice and humans that focus on EEC distribution, expression and hormone bioavailability. They have done a very good job of this. Importantly, the authors highlight compelling data challenging the canonical view of EECs being a one hormone-one EEC subtype, which opens many new therapeutic opportunities and ways in which to approach future EEC studies. In the last section (#5), they go on to discuss the role of the gut microbiota on altering EEC and gut barrier function. Lengthening this last section may be necessary as the focus of this review is EECs and gut barrier integrity, which in the current format, seems to be overshadowed by the preceeding sections. One primary suggestion is to include more literature on GLP-2 biology and its effects on gut barrier integrity, and any interactions between EECs and intestinal immune cells. 

Overall, this is a well-written review and inclusion of or re-angling of EECs with gut barrier function would fit the title and focus of this manuscript.

Minor comments include:

  1. Line 49: "thin" may be more appropriate than "narrow"
  2. Line 52: "villus tip where THEY", not "there"
  3. Line 55: "B - Schematic demonstrating that under the Wnt signal" may be more direct.
  4. Line 57: "Atoh1", not Atho1
  5. Line 74: "EECs are derived"
  6. Lines 290-293 are unclear. The compare and contrast of obesity effects on GLP-1 secretion is not clear.
  7. Line 454: not clear "which" refers to - oral vancomycin or Firmicutes?
